

# The role of macrophages in renal fibrosis and therapeutic prospects

Di Niu[1,*], Jun Jie Yang[2,*] and Dan Feng He[2]

[1] Department of Otolaryngology, The Eighth Hospital of Wuhan, Wuhan, China
[2] Department of Pharmacy, The Eighth Hospital of Wuhan, Wuhan, China
[*] These authors contributed equally to this work.

## ABSTRACT

Monocytes/macrophages are the key regulators of tissue repair, regeneration, and fibrosis. Monocyte-derived macrophages, which are characterized by high heterogeneity and plasticity, are recruited, activated, and polarized throughout the process of renal fibrosis in response to the local microenvironment. Increasing evidence suggests that phenotypic changes in macrophages are essential for chronic kidney disease (CKD) development and progression. Advanced bioinformatics and single-cell RNA sequencing analyses have revealed the critical mechanisms of macrophage iron homeostasis dysregulation and macrophage-to-myofibroblast transition (MMT), which may be a novel therapeutic target for renal fibrosis. In this review, we systematically examine the dynamic phenotype transitions of macrophages across distinct phases of kidney injury progression. Notably, we provide new insights into the multifaceted crosstalk between renal macrophages and neighboring parenchymal cells, including tubular epithelial cells, fibroblasts, podocytes, mesangial cells, and endothelial cells, mediated through diverse mechanisms, including soluble factors, extracellular vesicles, and direct cell-cell contact, and highlight the therapeutic potential of targeting macrophages.

## INTRODUCTION

The definition of chronic kidney disease (CKD) refers to chronic structural and functional abnormalities of the kidneys caused by various factors (with a history of kidney damage lasting more than 3 months) (*Jadoul, Aoun & Masimango Imani, 2024*; *Kidney Disease: Improving Global Outcomes CKDWG. KDIGO, 2024*). This condition includes pathological damage with or without abnormal glomerular filtration rate (GFR), abnormalities in blood or urine composition, imaging abnormalities, or an unexplained decline in GFR ($<60$ mL/min 1.73 m$^2$) persisting for more than 3 months (*Kidney Disease: Improving Global Outcomes CKDWG. KDIGO, 2024*; *Naber & Purohit, 2021*; *Collaboration GBDCKD, 2020*). In recent years, with the increasing aging population and increasing number of patients with diabetes, obesity, and hypertension, the prevalence of CKD has shown an annual upward trend. According to incomplete statistics, approximately 850 million people worldwide are affected by CKD (*Chen & Abramowitz, 2019*; *Kalantar-Zadeh & Li, 2020*). If CKD is not promptly and effectively treated, it can ultimately progress to

Corresponding author
Dan Feng He, rain251@126.com

end-stage renal disease (ESRD), necessitating long-term renal replacement therapy or kidney transplantation (*Ryu et al., 2022*; *Romagnani et al., 2025*).

Renal fibrosis is a common feature of CKD and a critical pathogenic factor that leads to ESRD (*Meng, Nikolic-Paterson & Lan, 2016*; *Liu, 2011*; *Shi et al., 2023*). Increasing evidence indicates that progressive renal fibrosis involves multiple contributing factors, including classical risk factors, microvascular damage, and inflammation. These pathological processes are closely associated with metabolic alterations mediated by hyperactive renin-angiotensin system, dysregulated aryl hydrocarbon receptor (AHR) signaling, aberrant Wnt/β-catenin and TGF-β/Smad pathways, as well as disturbances in endogenous metabolite homeostasis and the microbiome (*Huang, Fu & Ma, 2023*; *Krukowski et al., 2023*; *Ravid, Kamel & Chitalia, 2021*). A microbiome study comparing healthy controls and patients with CKD (stages 1–5) revealed that the abundance of *Lactobacillus johnsonii* was closely associated with clinical renal markers. Targeting *L. johnsonii* ameliorates membranous nephropathy by suppressing the aryl hydrocarbon receptor (AHR) signaling pathway (*Miao et al., 2024*). Epigenetic alterations play a crucial role in renal fibrosis development. Histone crotonylation (H3K9cr), which is significantly upregulated during renal fibrosis, promotes macrophage activation and tubular cell injury, exacerbating renal tissue fibrosis (*Li et al., 2024b*). Additionally, Sirtuin 6 attenuates renal fibrosis by epigenetically inhibiting the Wnt1/β-catenin pathway (*Cai et al., 2020*).

The progression of renal fibrosis is associated with various cellular activities, including dysregulated extracellular matrix remodeling, epithelial-mesenchymal transition of renal tubular cells, activation of mesangial cells and fibroblasts, inflammatory cell infiltration, and apoptosis (*Shi et al., 2023*; *Chen et al., 2021*). Multiple cell types, including immune cells, tubular epithelial cells (TECs), myofibroblasts, and podocytes, play a role in CKD development and contribute to progressive renal fibrosis (*Yamashita & Kramann, 2024*; *Sun et al., 2022*; *Zhang et al., 2023*). Immune cells, particularly macrophages, play crucial roles in CKD (*Abbad, Esteve & Chatziantoniou, 2025*). High densities of CD163[+] macrophages in kidney tissues are associated with poor renal function and an increased risk of ESRD (*Pfenning et al., 2023*).

Macrophages are highly heterogeneous plastic immune cells that are recruited, activated, and polarized in response to local microenvironmental signals during kidney injury. Their biological functions, ranging from promoting renal tissue damage to facilitating repair or driving fibrosis, are critically determined by their phenotypic polarization states and surrounding microenvironment (*Yonemoto et al., 2006*; *Calle & Hotter, 2020*; *Novak & Koh, 2013*).

Macrophages promote the pathogenesis of renal fibrosis by establishing intricate interaction networks with key cell types, including tubular cells, fibroblasts, and endothelial cells through direct interactions and/or secretion of soluble molecules such as hormones, growth factors, and cytokines (*Tian et al., 2024*; *Hoeft et al., 2023*). Advanced bioinformatics and single-cell RNA sequencing analyses have revealed critical mechanisms by which the macrophage-to-myofibroblast transition and dysregulation of macrophage iron homeostasis promote renal fibrosis. These findings may reveal novel therapeutic targets for renal fibrosis (*Wu et al., 2024*; *Chen et al., 2022*).

This review systematically elucidates the pivotal roles of macrophages in patients with CKD and animal models. We comprehensively summarize the multifunctional effects of dynamic macrophage phenotypic changes during CKD progression, including their intricate interactions with other cells in the renal microenvironment and their profound impact on cellular metabolic reprogramming and inflammatory regulation networks. Furthermore, we explored novel therapeutic strategies targeting the phenotypic modulation of macrophages in CKD.

## THE ROLE OF MACROPHAGES IN RENAL FIBROSIS

Macrophages are the primary immune cell population in normal kidneys and are regarded as crucial sentinels that play vital roles in the establishment and pathogenesis of acute kidney injury (*Meng, Jin & Lan, 2022*; *Privratsky et al., 2023*). Because of their diverse polarization states, macrophages infiltrating the kidneys exert profound effects on renal injury, repair, and fibrosis (*Cohen et al., 2024*; *Niculae et al., 2023*). Different macrophage subtypes are involved in the various stages of CKD. Macrophages exhibit high plasticity, allowing them to evolve into multiple phenotypes based on their microenvironment, thereby playing distinct roles in processes such as kidney injury, repair, or fibrosis (*Patino et al., 2023*; *Li et al., 2024a*).

### Macrophage origins and phenotypic diversity

Renal macrophages encompass resident and infiltrating cell populations. Resident macrophages in the kidney originate from diverse sources, including yolk sac-derived erythromyeloid progenitors (EMPs), fetal liver EMP-derived macrophages, and hematopoietic stem cells (HSCs)-derived macrophages (*Ma et al., 2022*; *Chen, Liu & Zhuang, 2022*). Notably, a subset of HSCs migrates to the bone marrow and spleen and subsequently differentiates into circulating monocytes that enter the bloodstream. These monocytes contribute to the pool of tissue-resident macrophages within the kidneys (*Cheung, Agarwal & George, 2022*; *Cheung et al., 2022*).

Macrophages can be classified based on their origin into bone marrow-derived macrophages and tissue-resident macrophages, and based on their function, activation state, and secreted factors, are categorized into M1 and M2 types (according to the level of lymphocyte antigen 6C [Ly6C]). Furthermore, these types can be divided into three subtypes: CD11b+/Ly6Chigh, CD11b+/Ly6Cintermediate, and CD11b+/Ly6Clow (Table 1) (*Fu et al., 2022*; *Duffield, 2011*).

M1 and M2 macrophages play opposing roles in renal inflammation (Fig. 1). During the initial phase of kidney injury, macrophages are activated by pathogen-associated molecular patterns (PAMPs), damage-associated molecular patterns (DAMPs), interferon-gamma (IFN-γ), and pro-inflammatory cytokines such as interleukin (IL)-1 and tumor necrosis factor (TNF)-α (*Cho et al., 2014*; *You et al., 2022*; *Fu et al., 2023*; *Zhou et al., 2019*; *Zhang et al., 2012*). This activation drives their differentiation into proinflammatory M1 macrophages, which respond to infections or cellular damage. Simultaneously, circulating monocytes (CD11b+Ly6Chigh) are recruited to the kidney, where they differentiate into proinflammatory M1 macrophages. Proinflammatory macrophages are the first responders

**Table 1  Mouse monocyte/macrophage markers.**

| Phenotype | Stimulant | Markers |
| --- | --- | --- |
| M1 | IFN-γ, LPS, GM-CSF, TNF-α | CXCL9, IL-12$^{high}$/IL-10$^{low}$, iNOS, IL-6, CD80, CD86, TNF-α |
| M2a | IL-4, IL-13 | CCL17, IL-1R, Dectin-1, IL-10, Arg-1, Chil3, FIZZ1 |
| M2b | LPS+IC, IL-1β+IC | CCL1, IL-10$^{high}$/IL-12$^{low}$, TNF-α, CD86, IL-6, LIGHT |
| M2c | IL-10, Glucocorticoids | CXCL13, CD206, CD163, IL-10, TGF-β, MerTK |
| M2d | LPS+A2R ligands, IL-6 | VEGF, IL-10, TGF-β, iNOS |

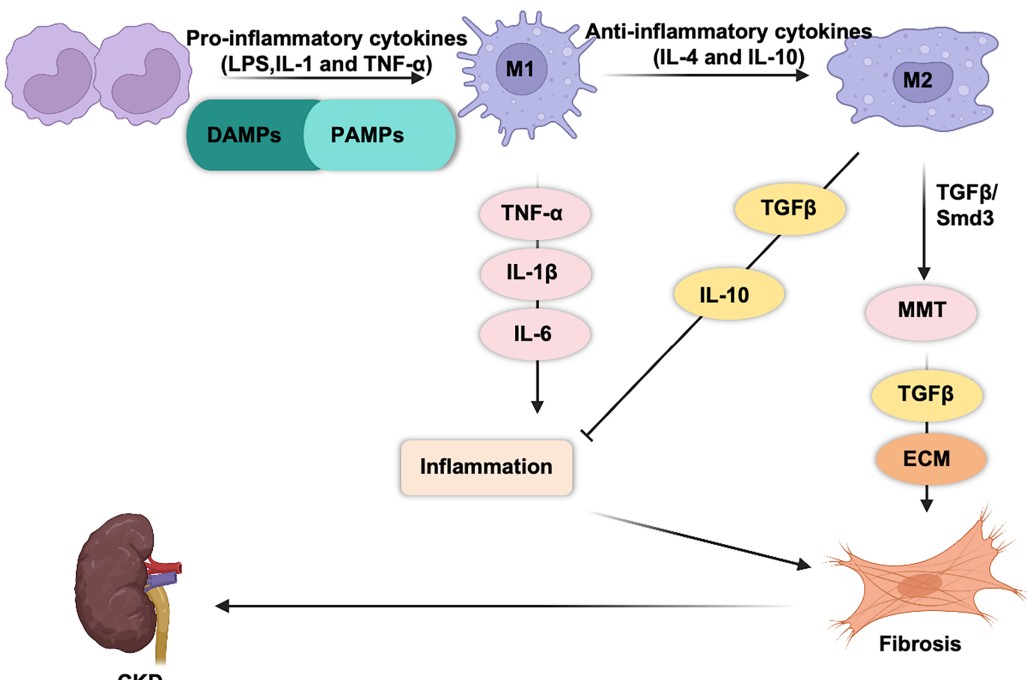

**Figure 1  Macrophages are generally classified into classic M1 and alternative M2 macrophages.** M1 macrophages express and secrete inflammatory cytokines (TNF-α, IL-1b, IL-6, IL-23, CCL2…) and promote tissue inflammation. M2 macrophages secrete anti-inflammatory (IL-10) and pro-fibrotic cytokines (TGF-β) that promote tissue repair and fibrosis.

to injury. These macrophages engulf cellular debris and secrete cytotoxic agents such as inducible nitric oxide synthase (iNOS) and reactive oxygen species (ROS) (*Fang et al., 2021*; *Tran & Mills, 2024*; *West et al., 2011*), both of which can induce mitochondrial damage and cell apoptosis (*Alexander et al., 2020*). M1 macrophages release pro-inflammatory cytokines (TNF-α, IL-1β, and IL-6), which exacerbate tissue inflammation and injury (*Hirani et al., 2022*; *Beyranvand Nejad et al., 2021*; *Tang, Nikolic-Paterson & Lan, 2019*).

M2 macrophages play a dual role in kidney repair. M2 macrophages secrete a variety of pro-fibrotic factors (such as TGF-β1, FGF-2, and PDGF) (*Luo et al., 2023*), which stimulate

the activation and proliferation of myofibroblasts and lead to excessive deposition of collagen in the extracellular matrix (ECM), thereby promoting fibrosis. In contrast, they release anti-inflammatory cytokines, such as IL-10, which help suppress renal inflammation (*Sun et al., 2024b*; *Liao et al., 2023*).

M2 macrophages can be further categorized into four subtypes: M2a, M2b, M2c, and M2d (*Yao, Xu & Jin, 2019*; *Shapouri-Moghaddam et al., 2018*). Among these, M2a macrophages express high levels of CD206, which are induced by IL-4 and IL-13 (*Luo et al., 2023*; *Chen et al., 2019*; *Rao et al., 2021*). M2a macrophages secrete pro-fibrotic mediators, including transforming growth factor-β (TGF-β), insulin-like growth factor (IGF), and fibronectin, thereby to contribute to tissue repair and wound healing (*Sindrilaru & Scharffetter-Kochanek, 2013*). M2b macrophages are induced by dual activation of Toll-like receptor (TLRs) or IL-1 receptor (IL-1R) ligands. This subtype plays a key role in immunoregulation and promotes T helper 2 (Th2)-like activation (*Wang et al., 2019*). M2c macrophages are polarized by IL-10, TGF-β, and glucocorticoids. This subtype exhibits potent immunosuppressive properties while actively participating in extracellular matrix remodeling and facilitating tissue repair (*Lurier et al., 2017*). M2d macrophages, which are activated by TLR ligands and A2 adenosine receptor agonists, play a pivotal role in regulating tumor progression, angiogenesis, and metastasis (*Hao et al., 2012*; *Li et al., 2023a*).

## The role of macrophages in renal fibrosis

Anti-inflammatory macrophages facilitate tubular reepithelialization by secreting trophic factors. However, persistent or severe inflammatory responses can also trigger renal fibrosis. Macrophage depletion alleviates renal fibrosis, highlighting the pro-fibrotic role of macrophages in various kidney pathologies (*Cohen et al., 2024*; *Ma et al., 2022*).

Recent single-cell sequencing studies have shown that macrophage infiltration increases significantly on the first day after ischemia/reperfusion, reaching a second peak by the 14th day. Spatial transcriptomic analysis revealed that during the early stages of AKI, macrophages extensively infiltrate the corticomedullary junction region, where tubular damage is severe, whereas in the chronic phase, they exhibit spatial proximity to fibroblasts (*Zhang et al., 2024b*).

Further, pseudotime analysis has identified two distinct macrophage lineages during the transition from AKI to CKD: kidney-resident macrophages can self-renew and differentiate into pro-repair subtypes, whereas monocyte-derived macrophages are involved in chronic renal inflammation and fibrosis. A novel subset of monocyte-derived macrophages, EAMs, has been identified and characterized by their ability to promote extracellular matrix remodeling. EAMs infiltrate the kidneys during the early stages of AKI and persist during the fibrotic phase. This subset facilitates chronic renal inflammation and fibrosis through intercellular communication with fibroblasts *via* an insulin-like growth factor (IGF) signaling pathway (*Zhang et al., 2024b*; *Du et al., 2019*).

Single-cell sequencing analysis of left ventricular myocardial infarction in mice revealed a distinct population of macrophages characterized by expression of Spp1, Fn1, and Arg1. This macrophage subset is upregulated in response to injury not only in the heart, but also

in other organs, including the kidney, where it actively promotes the progression of fibrotic processes (*Hoeft et al., 2023*). Differential gene expression analysis demonstrated that the most significantly altered factor in Spp1+ macrophages was chemokine ligand 4 (CXCL4). Genetic ablation of CXCL4 suppresses the activation of this macrophage population and attenuates fibrosis following renal injury (*Yang et al., 2022*).

Increased macrophage infiltration during renal aging leads to chronic low-grade inflammation, with enhanced communication between macrophages and renal tubular cells (*Li et al., 2024b*; *Sun et al., 2024a*). Ferroptosis inhibition can alleviate macrophage-mediated partial epithelial-mesenchymal transition in the renal tubules *in vitro*, thereby reducing the expression of fibrosis-related genes (*Cheng et al., 2023*). The natural small-molecule compound, rutin, which can inhibit macrophage senescence and ferroptosis by preserving Pcbp1, is a potential therapeutic agent for mitigating age-related chronic low-grade inflammation and fibrosis in kidneys (*Wu et al., 2024*). In addition, an interaction between hematopoietic cell kinase (HCK), an Src family kinase member, and ATG2A and CBL (two autophagy-related proteins) was found in unilateral ureteral obstruction and unilateral ureteral obstruction (UUO) models. Macrophage activation induced by autophagy inhibition promotes renal fibrosis (*Chen et al., 2023*).

Single-cell RNA sequencing analysis of a murine glomerulonephritis model revealed that VISTA-positive macrophages exert protective effects against glomerulonephritis and subsequent renal fibrosis. Mechanistically, these macrophages attenuate disease progression by suppressing IFN-$\gamma$ production in infiltrating T cells, which in turn reduces IL-9 cytokine production in downstream parenchymal response cells, ultimately alleviating glomerulonephritis-associated renal fibrosis (*Kim et al., 2022*). Recent studies have provided compelling evidence of the pivotal role of macrophages in diabetic nephropathy progression. Pathological analysis of renal autopsy specimens from 88 patients with type 2 diabetes and confirmed diabetic nephropathy revealed significant correlations; glomerular CD163+ macrophage infiltration was associated with disease severity, including interstitial fibrosis, tubular atrophy, and glomerulosclerosis, whereas interstitial CD68+ macrophages showed an inverse correlation with glomerular filtration rate and a positive association with albuminuria (*Klessens et al., 2017*). Advancements in single-cell RNA sequencing (scRNA-seq) technology have further elucidated macrophage dynamics in diabetic kidneys. Analysis of streptozotocin-induced diabetic mice demonstrated increased immune cell infiltration into glomeruli, with macrophage populations being particularly prominent (*Fu et al., 2019*). Furthermore, longitudinal single-cell transcriptomic profiling of CD45+ kidney immune cells in OVE26 type 1 diabetic mice revealed the temporal expansion of macrophage subsets accompanied by upregulated pro-inflammatory gene signatures (*Fu et al., 2022*).

## MACROPHAGE AND RENAL CELL CROSSTALK IN RENAL FIBROSIS

Macrophage infiltration in both the glomerular and interstitial compartments is a hallmark of CKD pathogenesis. The complex crosstalk between infiltrating macrophages and resident

renal cells, including tubular epithelial cells, fibroblasts, podocytes, endothelial cells, and mesangial cells plays a pivotal role in disease progression. Following renal injury, these renal cells secrete cytokines to communicate with each other, creating a pro-fibrotic microenvironment that drives disease advancement.

## The relationship between macrophages and epithelial cells

Thymic epithelial cells (TECs) or invading pathogens initiate immune responses by releasing damage-associated molecular patterns (DAMPs) and pathogen-associated molecular patterns (PAMPs). These molecular signatures are primarily detected in tissue-resident macrophages, which serve as the first line of immune surveillance. Macrophages recognize danger signals through their pattern recognition receptors (PRRs) and initiate a cytokine cascade. This inflammatory response not only activates local macrophages but also recruits additional monocyte-derived macrophages from the bone marrow, amplifying the immune reaction.

Macrophage-derived exosomes carrying miR-155 can be internalized by TECs, where the microRNA exerts its regulatory function by binding to the 3′-UTR region of TRF1. This interaction leads to downregulation of TRF1 expression, subsequently promoting telomere shortening and functional impairment in TECs. The resulting cellular senescence of TECs and exacerbation of renal fibrosis highlights the critical role of this macrophage-mediated exosomal pathway in kidney pathology (*Yin et al., 2024*). FKBP5 deficiency in TECs can reduce apoptosis, promote the proliferation of TECs, and inhibit M1 polarization and chemotaxis of macrophages to alleviate CaOx kidney stone injury in mice (*Song et al., 2023*).

Macrophage infiltration around lipotoxic TECs is a hallmark of diabetic nephropathy. Despite their clinical significance, the precise molecular mechanisms governing bidirectional communication between these cell populations remain poorly understood. Recent research has revealed that TECs can release specialized extracellular vesicles (EVs) enriched with LGR4, which serve as potent activators of macrophage functions. Conversely, activated macrophages can reciprocate by secreting distinct EV populations that initiate apoptotic pathways in the TECs, thereby establishing a vicious cycle of renal injury (*Jiang et al., 2022*).

In a murine model of adriamycin (ADR)-induced chronic proteinuric nephropathy, exosomal miR-19b-3p was derived from injured tubular epithelial cells. TECs are critical mediators of intercellular communication between damaged TECs and resident macrophages. This exosome-mediated signaling pathway specifically promotes polarization and activation of M1 macrophages, exacerbating renal inflammation and tissue injury (*Lv et al., 2020*).

## The relationship between macrophages and fibroblasts

The pathogenesis of renal fibrosis is characterized by the activation, proliferation, and phenotypic transformation of multiple renal cell types, including resident fibroblasts, epithelial cells, podocytes, and macrophages, into myofibroblasts. These activated myofibroblasts exhibit a distinctive secretory profile, characterized by the excessive

production of α-smooth muscle actin (α-SMA), various collagen subtypes, vimentin, platelet-derived growth factor receptor-β (PDGFR-β), and fibroblast-specific protein 1 (FSP-1). This aberrant extracellular matrix deposition and growth factor signaling creates a self-perpetuating fibrotic microenvironment that progressively compromises renal architecture and function.

Emerging evidence from recent investigations has fundamentally reshaped our understanding of the pathogenesis of renal fibrosis, highlighting the pivotal role of macrophages in CKD progression. Notably, lineage-tracing studies have demonstrated that bone marrow-derived macrophages can undergo phenotypic transformation into myofibroblasts through a well-defined process called macrophage-to-myofibroblast transition (MMT) (*Chen et al., 2022*; *Zhuang et al., 2024*; *Wang et al., 2017*). This cellular trans-differentiation process contributes to pathological extracellular matrix (ECM) deposition and serves as a key driver of fibrotic progression in various CKD models (*Chen et al., 2022*; *Wei, Xu & Yan, 2022*).

Investigations into MMT have primarily focused on characterizing intermediate transitional cell populations that exhibit dual expression of both myofibroblast and macrophage markers, specifically α-SMA and CD68. Notably, these α-SMA+CD68+ double-positive cells within fibrotic lesions co-express CD206, a well-established marker of M2 macrophage polarization, suggests a potential preferential involvement of M2-like macrophages in the MMT process (*Wang et al., 2022*; *Han et al., 2022*). In UUO mouse models, sophisticated lineage-tracing studies utilizing fluorescent protein labeling techniques have demonstrated that approximately 50% of α-smooth muscle actin-positive (α-SMA+) myofibroblasts are derived from myeloid lineage cells. Notably, a significant subset of these α-SMA+CD68+ transitional cells co-express CD206, a characteristic marker of M2 macrophage polarization (*Meng et al., 2016*). These findings suggest a potential differentiation cascade in which circulating myeloid cells migrate to sites of renal injury, initially adopting an M2-like macrophage phenotype before ultimately undergoing MMT to become fully differentiated myofibroblasts. This proposed multistep differentiation pathway (involving sequential phenotypic transitions from myeloid cells to M2 macrophages and finally to myofibroblasts) may represent a critical mechanism driving fibrotic progression in renal disease (*Wang et al., 2017*).

The estrogen membrane receptor, GPER1, alleviates the progression of renal fibrosis by inhibiting macrophage polarization towards both M1 and M2 phenotypes. GPER1 activation in M1 macrophages suppresses inflammatory signaling pathways, thereby protecting tubular epithelial cells (TECs) from immune-mediated activation and injury. Furthermore, GPER1 activation in M2 macrophages inhibits the transformation of resident fibroblasts into myofibroblasts, ultimately mitigating renal fibrosis (*Xie et al., 2023*).

Epigenetic modifications play a pivotal role in fibrotic progression in kidney diseases. Specifically, METTL3 promotes M2 macrophage-driven macrophage-to-myofibroblast transition (MMT) *via* modulation of the TGF-β1/Smad3 pathway, thereby exacerbating renal fibrosis in chronic renal transplant rejection (*Yao et al., 2025*). Macrophage Dectin-1 exacerbates Ang II-induced renal fibrosis by modulating TGF-β1 production and mediating macrophage-renal fibroblast interactions (*Ye et al., 2023*). Endogenous metabolites

 

promote tubulointerstitial fibrosis progression through aryl hydrocarbon receptor (AhR) signaling. The metabolite 1-methoxypyrene (MP) is a crucial metabolite that activates the AhR signaling pathway, thereby promoting tubulointerstitial fibrosis through both epithelial-mesenchymal transition and macrophage-myofibroblast transition (*Cao et al., 2022*). Consequently, targeting the cellular origins of myofibroblasts and elucidating the molecular mechanisms governing MMT have emerged as promising therapeutic strategies for modulating fibrotic progression (*Yuan et al., 2023*; *Xu et al., 2023*; *Zeng et al., 2022*).

## The relationship between macrophages and podocyte

Podocytes are highly specialized, terminally differentiated glomerular epithelial cells that play a crucial role in maintaining the kidney filtration barrier (*Nagata, 2016*). In diabetic nephropathy (DN), activated macrophages contribute to podocyte injury through multiple mechanisms including apoptosis induction and cytoskeletal disruption. Podocyte depletion directly correlates with the development and progression of proteinuria, a hallmark clinical manifestation of DN. Given their limited regenerative capacity, the degree of podocyte injury serves as a critical prognostic indicator to assess disease severity and predict renal outcomes in patients with DN (*Li et al., 2023b*; *Barutta, Bellini & Gruden, 2022*).

Single-cell RNA sequencing (scRNA-seq) analysis has demonstrated significant upregulation of TNF-α signaling pathways in human podocytes following growth hormone stimulation (*Wu et al., 2019*). Conditioned media collected from growth hormone-treated podocytes potently induced monocyte-to-macrophage differentiation. Notably, this differentiation capacity was substantially attenuated when the media were pretreated with a TNF-α neutralizing antibody, suggesting a TNF-α-dependent mechanism. *In vivo* studies using growth hormone-treated mice have revealed a pathological triad of increased macrophage infiltration, podocyte injury, and proteinuria development, further supporting the critical role of this signaling axis in glomerular pathology. Podocytes exposed to high-glucose culture conditions demonstrate a marked increase in their capacity to induce macrophage migration, with approximately 2.5-fold higher chemotactic activity than that of podocytes maintained under normal glucose conditions (*You et al., 2013*). Under diabetic conditions, podocytes undergo significant phenotypic alterations, leading to the sustained release of pro-inflammatory mediators including TNF-α and monocyte chemoattractant protein-1 (MCP-1). These cytokines orchestrate a complex inflammatory cascade by promoting macrophage chemotaxis and upregulating T-cell immunoglobulin and mucin domain-3 (TIM-3) expression in renal macrophages through activation of the NF-κB/TNF-α signaling pathway. M1 macrophage-derived exosomal miR-21a-5p and miR-25-3p promote podocyte apoptosis by directly targeting Tnpo1 and Atxn3, respectively (*Zhuang et al., 2022*). This self-perpetuating inflammatory cycle exacerbates podocyte injury and drives the progression of diabetic nephropathy, ultimately contributing to the deterioration of renal function in diabetic kidney disease (*Yang et al., 2019*).

Persistently high proteinuria levels are associated with poor long-term renal outcomes in patients with lupus nephritis (LN). Podocyte injury is responsible for substantial proteinuria. Thus, podocytes are key targets for LN therapy. Piezo1 knockout significantly reduced glomerulonephritis, tubulointerstitial injury, and podocyte foot process fusion,

and improved renal function and proteinuria in MRL/lpr mice (*Fu et al., 2024*). In IgAN, dysglycosylated IgA1 induces NLRP3 expression in podocytes, initiates podocyte macrophage transdifferentiation, and contributes to the inflammatory cascade and renal fibrosis changes associated with IgAN (*Qi, 2024*).

## The relationship between macrophages and mesangial cells

Mesangial cells, derived from mesenchymal stromal progenitors, form a critical structural and functional component of the glomerular tuft, where they interact with the mesangial matrix to maintain the glomerular architecture and regulate the glomerular microvascular network. These specialized perivascular cells play a central role in glomerular homeostasis and their dysfunction contributes significantly to the pathogenesis of various glomerular diseases. Upon pathological stimulation, mesangial cells are activated through the MAPK and PKC signaling cascades, triggering a series of molecular events that drive both inflammatory responses and fibrotic processes within the glomerulus (*Zhao, 2019*; *Hu et al., 2024*). Injured mesangial cells actively recruit circulating monocytes and macrophages to the sites of glomerular damage through the secretion of chemotactic factors. Notably, these infiltrating macrophages predominantly exhibit an M1-polarized phenotype, a phenomenon potentially mediated by intricate crosstalk between the NOTCH signaling pathway and NF-κB activation, which collectively establishes a proinflammatory microenvironment within the glomerulus (*Ma et al., 2022*).

Exosomes serve as crucial mediators of intercellular communication between macrophages and mesangial cells in the renal microenvironment (*Zhu et al., 2020*; *Phu et al., 2022*). Specifically, exosomes secreted by macrophages exposed to high glucose conditions have been shown to potently induce mesangial cell activation and proliferation, leading to pathological mesangial expansion and the subsequent secretion of pro-inflammatory cytokines. Mechanistic studies have revealed that these macrophage-derived exosomes contain elevated levels of TGF-β1, which activates the TGF-β1/Smad3 signaling cascade in recipient mesangial cells, ultimately driving excessive extracellular matrix deposition in both *in vitro* and *in vivo* models of diabetic nephropathy (*Zhu et al., 2020*). Macrophages regulate mesangial cell proliferation and migration through CXC motif chemokine ligand 12 (CXCL12)/dipeptidyl peptidase 4 (DPP4) axis interaction, leading to disruption of the LN filtration barrier and impaired renal function (*Li et al., 2024c*).

## The relationship between macrophages and endothelial cells

As integral components of the glomerular filtration barrier, glomerular endothelial cells maintain a direct interface with the circulatory system, making them particularly vulnerable to physiological and pathological blood-borne factors (*Zhang et al., 2024a*). Renal-resident macrophages are predominantly localized in the periglomerular interstitial space and are strategically positioned to monitor glomerular function. The pro-inflammatory cytokine, TNF-α, upregulates endothelial adhesion molecule expression, thereby promoting leukocyte extravasation and subsequent infiltration into renal parenchymal tissue (*Summers et al., 2011*).

In a recent study on septic mice, the authors utilized single-cell sequencing revealed that F4/80hi macrophages undergo the most significant changes, with a notable reduction in the

proportion of this macrophage subset following sepsis. Compared to other macrophage subsets, F4/80hi macrophages highly express anti-inflammatory genes, such as *Socs3*, *Il1r2*, and *Il1rn*. Ablation of F4/80hi macrophages exacerbated sepsis-induced AKI. Mechanistically, F4/80hi macrophages inhibit endothelial cell expression of IL-6 through IL1ra (an IL-1 receptor antagonist), ultimately mitigating sepsis-induced kidney damage (*Privratsky et al., 2023*).

Under hyperglycemic conditions, endothelial cells exhibit significant upregulation of the hypoxia-inducible factor-1α (HIF-1α)/Notch1 signaling pathway, which orchestrates the recruitment of pro-inflammatory M1 macrophages to renal tissue, thereby exacerbating renal injury in db/db diabetic mice (*Torres et al., 2020*). Therapeutic intervention with fenofibrate, a peroxisome proliferator-activated receptor alpha (PPAR-α) agonist, effectively attenuates this pathological process by suppressing HIF-1α/Notch1 signaling, resulting in reduced M1 macrophage infiltration and subsequent protection against diabetic nephropathy progression (*Torres et al., 2020*).

## Macrophage-renal cell crosstalk in AKI-to-CKD transition
### The crosstalk between renal macrophages and tubular epithelial cells
The CSF family is comprised of a group of cytokines involved in the differentiation and maturation of bone marrow cells in mammals. Injured proximal tubules are an important source of colony-stimulating factor 1. In a mouse model of ischemia-reperfusion-induced AKI, the tubule-specific conditional knockout of CSF-1 drives macrophage proliferation and M2 phenotype polarization, thereby promoting kidney recovery and reducing renal interstitial fibrosis (*Wang et al., 2015*). In addition, injured tubular cells secrete granulocyte-macrophage colony-stimulating factor (GM-CSF), which promotes monocyte/macrophage infiltration in a macrophage chemoattractant protein-1 (MCP-1)-dependent manner, leading to sustained inflammation and tubular apoptosis (*Xu, Sharkey & Cantley, 2019*).

Macrophage polarization is dynamically regulated by miRNA levels within the microenvironment (*Essandoh et al., 2016*). Notably, macrophage-derived exosomal miRNAs serve as key mediators of intercellular communication with renal cells by modulating critical signaling pathways. In the LPS-induced AKI murine model, we observed significant upregulation of exosomal miR-19b-3p in tubular epithelial cells (TECs), which promoted M1 macrophage polarization. Similarly, miR-374b-5p exhibits comparable pro-inflammatory effects. Mechanistic studies have revealed that both miRNAs activate NF-κB signaling by directly suppressing SOCS1 expression (*Lv et al., 2020*; *Ding et al., 2020*).

### Crosstalk between renal macrophages and fibroblasts
Recent studies have emphasized the role of platelet activation and platelet-derived factors in mediating macrophage-fibroblast crosstalk after kidney injury (*Jansen, Florquin & Roelofs, 2018*). In a murine model of ischemia-reperfusion injury (IRI)-induced AKI, integrated single-cell RNA sequencing (scRNA-seq) and spatial transcriptomics identified a distinct subset of macrophages, termed cycling M2 macrophages, which exhibited heightened proliferative activity. These macrophages, which are regulated by platelet-derived thrombospondin-1 (THBS1), adopt a profibrotic phenotype and frequently

interact with fibroblasts, particularly in the presence of platelets, during AKI-to-CKD progression. Treatment with a THBS1-blocking antibody (R300) markedly reduces the abundance of cycling M2 macrophages and downregulates fibroblast-expressing profibrotic genes linked to collagen synthesis and immune modulation (*Liu et al., 2024*).

### Crosstalk between renal macrophages and vascular endothelial cells

During injury and inflammation, endothelial cells serve as key regulators by secreting chemoattractants and expressing adhesion molecules, thereby facilitating leukocyte recruitment to sites of renal damage. Macrophages play a crucial immunomodulatory role in VECs, helping maintain tissue homeostasis post-injury. For example, vascular-resident CD169+ macrophages downregulate intercellular adhesion molecule-1 (ICAM-1) expression in VECs, thereby limiting excessive neutrophil infiltration and subsequent inflammatory cascades in mouse models of ischemia/reperfusion injury-induced AKI (*Karasawa et al., 2015*). Similarly, macrophage-expressed interleukin-1 receptor antagonist antagonize the IL-1 signaling pathway in endothelial cells, thereby suppressing IL-6 production and attenuating sepsis-induced renal inflammation in mice (*Privratsky et al., 2023*).

## THERAPEUTIC POTENTIALS OF MACROPHAGES IN CKD

Immune regulation plays a pivotal role in CKD pathogenesis and progression by orchestrating a complex cascade of pathological events. These include the recruitment and infiltration of diverse immune cell populations, sustained release of proinflammatory cytokines and chemokines, and deposition of immune complexes within the renal tissue, collectively contributing to the development of tubular and tubulointerstitial injury.

Recent single-cell RNA sequencing studies in a murine model of ischemia-reperfusion injury-induced renal fibrosis suggested that Fn1+Spp1+Mrc1+ macrophages may be a critical subset mediating renal fibrosis. To target this pro-fibrotic macrophage population, a novel bioactivatable *in vivo* assembly peptide (BIVA-PK) was developed. This innovative therapeutic agent specifically targets pro-fibrotic macrophages and undergoes enzyme-triggered self-assembly upon cleavage by cathepsin B, which is highly expressed in kidneys. Self-assembled nanostructures incorporate cell-penetrating peptides that induce selective macrophage apoptosis, thereby remodeling the renal immune microenvironment and effectively attenuating fibrosis progression (*Ouyang et al., 2024*).

Researchers developed a targeted nanoparticle system for the co-delivery of an endoplasmic reticulum stress (ERS) inhibitor (Ceapin 7) and conventional glucocorticoid (dexamethasone) to precisely modulate the ATF6/TGF-β/Smad3 signaling axis in macrophages. This innovative approach promotes macrophage polarization towards the M2c phenotype while suppressing excessive MMT, thereby effectively attenuating renal fibrosis progression (*Luo et al., 2023*).

In a nephrotoxic serum nephritis (NTN) model, myeloid-specific deletion of Krüppel-like factor 4 (KLF4) exacerbated both glomerular and tubular injury. Mechanistically, KLF4 attenuates renal inflammation and fibrosis by suppressing TNF-α production. Notably, the pharmacological inhibition of TNF-α receptor 1 significantly ameliorates renal fibrosis

and necrosis in mice with myeloid-specific KLF4 deficiency during NTN progression (*Wen et al., 2019*).

## LIMITATIONS

Current limitations in understanding the role of macrophages in renal fibrosis primarily include the following. First, while most studies rely on the M1/M2 dichotomy, macrophages in renal fibrosis likely exhibit a more complex phenotypic spectrum (*e.g.*, Fn1+Spp1+Mrc1+ pro-fibrotic subsets), with their polarization states dynamically changing across disease stages. Conventional techniques (*e.g.*, flow cytometry and immunohistochemistry) lack sufficient resolution to comprehensively capture this dynamic heterogeneity. Second, macrophage-targeted therapies (such as BIVA-PK nanoparticle-induced M2 apoptosis) may inadvertently affect other immune cell functions, potentially leading to immunosuppression.

## SUMMARY

In this comprehensive review, we systematically explore the multifaceted interplay between macrophages and renal fibrosis in the context of CKD progression. By integrating recent advances in single-cell omics and molecular biology, we aimed to provide a mechanistic understanding of how macrophage heterogeneity and plasticity contribute to the pathogenesis of CKD-related fibrosis and discuss emerging therapeutic strategies targeting macrophage-mediated pathways.

Although essential for tissue homeostasis, the process of renal injury repair paradoxically serves as the driving force for renal fibrosis. Within this context, the M2a and M2c macrophage subsets play crucial roles in facilitating MMT, a key pathological mechanism. Renal fibrosis primarily originates from the activation, proliferation, and phenotypic transformation of multiple renal cell types, including resident fibroblasts, epithelial cells, podocytes, and macrophages into ECM-producing myofibroblasts in response to renal injury. This cellular reprogramming leads to excessive ECM deposition and the progressive deterioration of renal function. MMT has emerged as a central mechanism of renal fibrosis.

We propose that MMT is a critical mechanistic link between renal inflammation and fibrosis. Although significant progress has been made in understanding MMT, several enigmatic aspects of this cellular transdifferentiation process remain to be elucidated, and additional molecular mechanisms underlying MMT regulation are likely to be discovered. From a therapeutic perspective, targeting the MMT pathway represents a promising and innovative strategy for developing novel interventions to prevent or attenuate the progression of renal fibrosis in CKD. Future research focusing on MMT modulation may pave the way for effective and precise antifibrotic therapies.

### Survey methodology

Literature searches were conducted using PubMed and Web of Science. In addition to articles published in 2019, earlier articles were considered. The following keywords were used: chronic kidney disease, macrophages, M1 and M2 phenotypes, interstitial fibrosis,

crosstalk, and therapeutic potential. As our work gradually unfolded, we then searched literature by keywords macrophages and renal cells, macrophages and epithelial cells, macrophages and fibroblasts, macrophages and podocyte, macrophages and mesangial cells, macrophages and endothelial cells, and therapeutic potentials of macrophages in CKD after removing duplicate articles and the articles with little relevance, 118 articles were selected for this review.

### Funding
The authors received no funding for this work.

### Competing Interests
The authors declare there are no competing interests.

### Author Contributions
- Di Niu conceived and designed the experiments, performed the experiments, analyzed the data, prepared figures and/or tables, authored or reviewed drafts of the article, and approved the final draft.
- Jun Jie Yang performed the experiments, prepared figures and/or tables, authored or reviewed drafts of the article, and approved the final draft.
- Dan Feng He conceived and designed the experiments, authored or reviewed drafts of the article, and approved the final draft.

### Data Availability
This is a literature review.

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
