# Peer review of "The role of macrophages in renal fibrosis and therapeutic prospects"

_PeerJ, doi:10.7717/peerj.19769_

## Round 0.1 · original submission · Major Revisions

**Language Note:** The review process has identified that the English language must be improved. PeerJ can provide language editing services - please contact us at [email protected] for pricing (be sure to provide your manuscript number and title). Alternatively, you should make your own arrangements to improve the language quality and provide details in your response letter. – PeerJ Staff

·

Basic reporting

In this work, the authors summarized the role of macrophages in renal fibrosis and therapeutic prospects. Several suggestions are made as follows to improve the quality of the manuscript.
1. The abstract and main text are two separate sections. All abbreviations should be substantiated for the first time in the main text.
2. In abstract, several statements are non-sensical. The abstract should be rewritten.
3. The latest studies uncovered the novel mechanism in renal fibrosis based on the renal tubular epithelial cells and podocytes. In the introduction, this reviewer recommends two general improvements; first, an opening statement reported novel mechanism of renal fibrosis such as the activation of renin-angiotensin system, Wnt1/β-catenin pathway and Sirtuin 6 as well as Lactobacillus johnsonii to reverse chronic kidney disease and Lactobacillus species ameliorate membranous nephropathy through inhibiting the aryl hydrocarbon receptor pathway would catch the reader’s interest. Secondly, the reviewer suggested that authors presented the novel mechanism of macrophages-mediated renal fibrosis.
4. The many descriptions need to provide the references such as “Studies have confirmed that multiple cell types…and contribute to progressive renal fibrosis.”
5. The authors summarized the role of macrophages in renal fibrosis. The latest studies have demonstrated that the macrophages mediated in renal fibrosis were associated with a variety of underlying molecular mechanisms reported by some publications such as Cellular and Molecular Life Sciences 2023;80(7):184, Integrative Medicine in Nephrology and Andrology 2023;10(3):e00001, Cellular and Molecular Life Sciences 2023;80(10):301, Acta Pharmacologica Sinica 2022;43(11):2929-2945. Please discuss these studies to improve manuscript.
6. The clinical studies should be discussed and improved.
7. The latest publications should be further discussed.
8. Limitations should be described.
9. Please change the references based on the guide for authors.
10. The language editing should be improved by a native speaker.

Experimental design

See the basic reporting.

Validity of the findings

See the basic reporting.

Additional comments

See the basic reporting.

·

Basic reporting

No comment

Experimental design

No comment

Validity of the findings

No comment

Additional comments

I suggest to associate fibrosis due to TGF-betha with protein loss and also miRNAs involved in those protein loss (https://doi.org/10.1016/j.genrep.2025.102173)

Reviewer 3 ·

Basic reporting

An interesting review article, which includes a sufficient number of original scientific papers to provide a comprehensive overview of the role of macrophages in renal fibrosis and therapeutic prospects.
Comments: The text contains minor typographical errors that should be corrected. Additionally, some parts of the text are missing references. In the introduction, it would be useful to add a few sentences regarding the etiological factors influencing the development of CKD.
Section 2.1: DAMPs are not "danger-associated" but "damage-associated molecular patterns." Also, in this section, it is necessary to specify which pro-inflammatory cytokines activate macrophages.
When looking at Figure 1, it appears that the end effect of M1 and M2 activation is the same, which is inconsistent with the previously presented text.
In Section 3.3, you examine the relationship between macrophages and podocytes, but only in the context of diabetic nephropathy. If available, please also include studies that have investigated this relationship in other kidney disorders. The same applies to Section 3.4.
It would also be beneficial to discuss the interactions between macrophages and podocytes, mesangial cells, and endothelial cells in acute kidney injury, considering its potential progression to CKD. Are similar mechanisms observed as in diabetic nephropathy?
In the conclusion, you mention M2A and M2C macrophages, which were not previously explained. It is necessary to clarify these subpopulations earlier in the text. Including a greater number of figures would certainly enrich your review article.

Experimental design

No comment.

Validity of the findings

No comment.

Additional comments

No comment.

---

## Round 0.2 · accepted · Accept

Dear Dr. He,

Thank you for submitting the revised version of your manuscript. After a thorough evaluation of your revisions by Reviewers and me, I am pleased to inform you that all reviewer comments have been satisfactorily addressed. Accordingly, your manuscript is now accepted for publication in PeerJ.

Sincerely,
Stefano Menini

·

Basic reporting

The authors have improved the manuscript, so I suggest that the manuscript should be accepted for publication.

Experimental design

-

Validity of the findings

-

·

Basic reporting

-

Experimental design

-

Validity of the findings

-

Additional comments

You may read this publication: https://doi.org/10.1016/j.genrep.2025.102173. TGF-beta linking to CKD progression in SRNS, which is the prototype, is nearly similar to diabetic kidney disease.

Reviewer 3 ·

Basic reporting

The authors have successfully responded to all the objections raised.

Experimental design

-

Validity of the findings

-